# A New Phenomenological Model for Single Particle Erosion of Plastic Materials

**DOI:** 10.3390/ma12010135

**Published:** 2019-01-03

**Authors:** Jiarui Cheng, Ningsheng Zhang, Liang Wei, Hongxue Mi, Yihua Dou

**Affiliations:** 1State Key Laboratory of Multiphase Flow in Power Engineering, Xi’an Jiaotong University, Xi’an 710049, China; nszhang@xsyu.edu.cn; 2Xibu Drilling Engineering Company Limited, China National Petroleum Corporation, Kelamayi 834000, China; xjsywl@sina.com (L.W.); xjsymhx@cnpc.com.cn (H.M.); 3Department of Mechanical Engineering, Xi’an Shiyou University, Xi’an 710065, China; yhdou@vip.sina.com

**Keywords:** single particle erosion, phenomenological model, Hertzian contact theory, critical sliding angle, gas-solid jet experiment

## Abstract

A phenomenological model for single particle erosion (SPE) of plastic materials was constructed based on the Hertzian contact theory and conservation of momentum to solve the particle impact erosion. The extrusion deformation and contact time of materials in three processes of wall elastic extrusion, elastic-plastic extrusion, and elastic recovery were discussed. Later, the critical angle for sliding contact between the particle and metal surface was calculated according to the impact angle of a particle and the corresponding critical sliding friction force of the particle. The wall indentation depths under sliding contact and no sliding contact were compared. Finally, the erosion volume of materials by impact of a single particle was gained. Moreover, a contrastive analysis on calculation results was carried out by using the gas-solid jet erosion experiment. Contact time, normal and tangential deformations of materials, as well as material erosion under sliding contact and no sliding contact in two processes of particle extrusion and rebound were gained from calculation and experiment. The constructed model showed a good agreement without involving too many empirical coefficients.

## 1. Introduction

The particles in liquid-solid or gas-solid two-phase flow are inevitable to impact the pipe wall, thus forming particle impact erosion and finally causing material loss on pipe walls. The erosion damage caused by particle impact can be relieved by improving the structure in accordance with thickness of the erosion wall. As a result, some studies interpreted the impact failure mechanism through an erosion experiment and investigated influences of particle impact velocity, impact angle, geometry of particles, material roughness, and hardness on erosion rate. The rest studies gained the erosion prediction method by building a particle impact model, thus enabling evaluation of the material loss on the wall surface.

The calculation of deformation and loss volume of metal surface caused by single particle impact involves studies concerning contact between particles and metal surface, deformation of metal surface, and particle deformation during 1960s~1970s [1,2,3,4]. Meng [5] summarized frequency of occurrence of different parameters in the erosion prediction model. He concluded that particle density, particle volume, particle impact velocity and angle, as well as surface hardness of target material occur mostly in the model. Finnie [6] proposed an erosion model which considered influences of kinetic energy of particles and material surface properties on erosion rates. Results demonstrated that loss of plastic materials was mainly caused by accumulation of plastic deformations, whereas loss of brittle materials was a consequence of crack development and breakage. This model proposed the division of critical particle impact angle creatively. When the impact angle was larger than this critical angle, the erosion rate and impact angle presented an increasing function relationship. Otherwise, they showed an inverse function relationship. Bitter [7] proposed the microcutting model which considered various material properties. In addition, the particle impact process was further refined into vertical indentation and horizontal cutting. Material loss volumes in these two processes were calculated. In the model proposed by Neilson and Gilchrist [8], the material loss smaller than the critical angle was expressed by the cutting formula, but the material loss larger than the critical angle was expressed by the indentation formula. Although the new formula was simpler than the Bitter’s formula, it had great errors because of independent considerations to indentation or cutting. Sheldon and Kanhere [9] constructed an erosion prediction model which didn’t divide by the impact angle. This model only involved some parameters which are easy to be gained, such as particle diameter, particle density, particle impact speed, and surface hardness of materials. Based on the low-cycle fatigue theory, Hutchings [10] created a material erosion model under continuous impact of particles. This model achieved high calculation accuracy under the impact angle of 90°, but it performed poorly under rest impact angles. Huang [11] set up a systematic particle impact-induced erosion prediction model. Based on conditions for conservation of kinetic energy during particle impact, this model divided the material deformation into tangential and vertical components. The vertical indentation and tangential extrusion were gained from a complicated physical deduction, which were used to form the total erosion rate. Moreover, this model also introduced in the low-cycle fatigue formula based on the material deformation caused by single particle impact to calculate material strain under the impacts of multiple particles. Finally, an erosion rate prediction model which was applicable to continuous impacts of particles was acquired.

Most recent research [12,13,14,15] focuses on numerical prediction of erosion rather than modelling. Although the macroscopic calculation of erosion of complicated structure was realized, there are still some problems in application of models. On the one hand, erosion models used in macroscopic calculation generally contain few physical parameters and replace rest necessary parameters by empirical coefficients gained from experiments. These models [6,7,8,9,16,17,18] depend highly on experimental measurement and have complex application steps, even though they show high prediction accuracy. On the other hand, phenomenological models with explicit physical significance were gained through complicated theoretical deduction. These models involve many physical parameters and assumed the particle–wall contact as a complete elastic impact, resulting in the great deviations between practical results and experimental results.

In this study, a physical model is created to study the single particle erosion of plastic materials. This model divided the particle–wall contact process into elastic indentation, elastic-plastic indentation and elastic recovery. By means of Hertzian contact theory and conservation of particle momentum, a surface erosion prediction model for single particle impacting plastic materials was obtained. Contact time and erosion volume of target materials under different component velocities of impact, particle diameters, particle density, and yield strengths of materials were calculated theoretically. Extrusion deformations under sliding contact and no sliding contact between a particle and the wall as well as the critical impact angle for sliding were calculated theoretically. The calculation results were compared with the results of gas-solid particle impact experiment. The calculated results showed a good agreement without involving too many empirical coefficients.

## 2. Model Establishment

Since no material is perfectly rigid, the imposition of a road always produces a deformation between the particle and wall in a collision. A typical particle–wall contact process can be divided into the following three steps:

Elastic compression: As shown in Figure 1a, when the contact center of wall is first deformed by a particle, the particle is decelerated by the normal contact force. At this time, the reduced kinetic energy of the particles is converted into wall and particle deformation elastic energy for subsequent energy release.

Elastic-plastic compression: Once the wall strain is greater than its yield limit, as shown in Figure 1b, a plastic deformation zone is first produced in the contact area. As the particles continue to compress, as shown in Figure 1c, the plastic deformation zone continues to expand toward the periphery of the contact area until the normal velocity of the particles drops to 0 m/s. The indentation strain of the wall reaches the maximum value.

Elastic recovery: When the particle velocity reaches the minimum, the elastic potential energy stored in the wall deformation zone is released, thereby pushing the particles in the opposite direction. In the release process, if the particle is not affected by external forces, theoretically it will always be in contact with the wall. However, if the particle is subjected to other contact forces (liquid forces, other particle collision forces, etc.), or non-contact forces (van der Waals forces, electrostatic forces, etc.), the particle and wall may come out of contact before the wall elastic deformation completely recovers.

In calculating the wall deformation process, the following assumptions are made:(1)The wall is made of isotropic material.(2)The particle is treated as fully rigid.(3)The particle is a sphere.(4)The vibration of the collision process is ignored.(5)The effect of temperature on the flow stress and yield stress of the extrusion process is ignored.

According to the previous research, a single particle extrusion geometry model is established as shown in Figure 2. The particle impact velocity is divided into normal and tangential to the wall.

### 2.1. Calculation of Normal Indentation

Based on Hertzian contact theory [20], The normal stress and deformation when the rigid particles are in contact with the wall surface can be expressed as,
(1)$$\sigma (r)=\sigma_{o}{\left(1-r^{2}/R^{2}\right)}^{1/2},$$
(2)$$\delta_{y}=\frac{\pi \sigma_{o}}{4E^{*}R}\left(2R^{2}-r^{2}\right)\;r\;≦\;R,$$ where 1/*E^*^* = (1 − *ν*^2^)/*E* + (1 − *ν_i_*^2^)/*E_i_*. According to the relationship between the maximum depth of elastic indentation *h*_1_ and normal deformation, one has
(3)$$\delta_{y}=h_{1}-r^{2}/2r_{p}^{},$$

Substituting Equation (2) into Equation (3), we have,
(4)$$\frac{\pi \sigma_{o}}{4E^{*}R}\left(2R^{2}-r^{2}\right)=h_{1}-r^{2}/2r_{p}^{}$$

Comparing the items in the polynomial, we get the expressions of *R* and *h*_1_,
(5)$$R=\frac{\pi \sigma_{o}r_{p}}{2E^{*}},$$
(6)$$h_{1}=\frac{\pi R\sigma_{o}}{2E^{*}}.$$

According to the geometric formula *R*^2^ = *r_p_* × *h*_1_, the maximum stress on the contact surface is
(7)$$\sigma_{o}=\frac{2E^{*}}{\pi }{\left(\frac{h_{1}}{r_{p}}\right)}^{1/2}.$$

And the total normal pressing force is expressed as,
(8)$$F_{y}=\int_{0}^{R}\sigma \left(r\right)2\pi rdr=\frac{2}{3}\sigma_{o}\pi R^{2}=\frac{4}{3}E^{*}r_{p}^{1/2}\delta_{y}^{3/2}.$$

The relationship between h_1_ and normal squeeze force is,
(9)$$F_{y,1}=\frac{4}{3}E^{*}r_{p}^{1/2}h_{1}^{3/2}.$$

By means of kinetic energy theorem, the motion equation of a particle during wall elastic deformation can be presented as,
(10)$$\frac{1}{2}m_{p}v_{y0}^{2}-\frac{1}{2}m_{p}v_{y1}^{2}=\int_{0}^{h_{1}}\frac{4}{3}E^{*}r_{p}^{1/2}\delta_{y}^{3/2}d\delta_{y}=\frac{8E^{*}r_{p}^{1/2}h_{1}^{5/2}}{15m_{p}},$$ where *v_y_*_0_ is the initial velocity since the particles are in contact with the surface; *v_y_*_1_ is the particle velocity at which the elastic indentation reaches maximum. Equation (10) can be recast as,
(11)$$v_{y1}^{}=\sqrt{v_{y0}^{2}-\frac{16E^{*}r_{p}^{1/2}h_{1}^{5/2}}{15m_{p}}}.$$

If we set the velocity *v_y_*_1_ to 0 m·s^−1^, it means that all the kinetic energy of the particles is converted into the elastic potential energy of the wall. Using the Equation (11) to calculate the initial particle velocity, which refers to the critical initial velocity (*v_y_*_0_) of a particle corresponding to the elastic-plastic deformation of the wall. If *v_y_* > *v_y_*_0_, the plastic deformation of wall will take place, otherwise, it will not occur. Therefore, we set *v_y_*_1_ to 0 m·s^−1^, the critical velocity *v_y_*_0_ can be presented as,
(12)$$v_{y0}^{}=\frac{16E^{*}r_{p}^{1/2}h_{1}^{5/2}}{15m_{p}}=0.19\times \frac{R^{5/2}r_{p}^{1/2}\sigma_{y}^{5/2}}{m_{p}{\left(E^{*}\right)}^{3/2}}.$$

According to the critical velocity of a particle before plastic deformation of the wall, the particle momentum equation is built to calculate the elastic contact time, which can be expressed by,
(13)$$\int_{0}^{h_{1}}F_{y}t_{1}d\delta_{y}=m_{p}(v_{y0}-v_{y1})=m_{p}(v_{y0}-\sqrt{v_{y0}^{2}-\frac{16E^{*}r_{p}^{1/2}h_{1}^{5/2}}{15m_{p}}}).$$

Substituting Equation (8) into Equation (13), it is recast as,
(14)$$\int_{0}^{h_{1}}\frac{4}{3}E^{*}r_{p}^{1/2}\delta_{y}^{3/2}t_{1}\cdot d\delta_{y}=m_{p}(v_{y0}-\sqrt{v_{y0}^{2}-\frac{16E^{*}r_{p}^{1/2}h_{1}^{5/2}}{15m_{p}}}).$$

Integrating Equation (14), the elastic contact time is,
(15)$$t_{1}=(v_{y0}-\sqrt{v_{y0}^{2}-\frac{16E^{*}r_{p}^{1/2}h_{1}^{5/2}}{15m_{p}}})\cdot \frac{15m_{p}}{8E^{*}r_{p}^{1/2}h_{1}^{5/2}}.$$

If 15*m_p_*/16*E^*^*·*r_p_*^1/2^·*h*_1_^5/2^ = *λ*, Equation (15) is recast as,
(16)$$t_{1}=m_{p}(v_{y0}-\sqrt{v_{y0}^{2}-1/\lambda })\cdot \lambda$$

In this case, the elastic contact time is dominated by the initial velocity of a particle and the maximum depth. When the wall is plastically strained, the contact stress should be equal to the yield stress of the material (i.e., *σ*_0_ = *σ_y_*_)_. The elastic contact time is expressed by *σ_y_* as shown below,
(17)$$t_{1}=(v_{y0}-\sqrt{v_{y0}^{2}-0.19\times \frac{R^{5/2}r_{p}^{1/2}\sigma_{y}^{5/2}}{m_{p}{\left(E^{*}\right)}^{3/2}}})\cdot 10.60\times \frac{m_{p}{\left(E^{*}\right)}^{3/2}}{R^{5/2}r_{p}^{1/2}\sigma_{y}^{5/2}}.$$

Similar to the calculation of elastic contact time of indention, using momentum theorem to establish a formula for calculating elastic-plastic contact time (*t*_2_) and rebound time (*t*_3_) between a particle and the wall. The specific process can refer to Appendix A. The elastic-plastic contact time is obtained after simplification, which is shown below,
(18)$$t_{2}=m_{p}\sqrt{v_{y0}^{2}-\frac{16E^{*}r_{p}^{1/2}h_{1}^{5/2}}{15m_{p}}}/\left(0.17\times \frac{\pi^{3}\sigma_{y}^{3}r_{p}^{2}h_{2}}{{\left(E^{*}\right)}^{2}}+\frac{1}{2}\sigma_{y}\pi r_{p}^{}h_{2}^{2}\right).$$

The calculation of rebound time can be expressed by,
(19)$$t_{3}=m_{p}\sqrt{16E^{*}r_{p}^{1/2}h_{1}^{5/2}/15m_{p}}\cdot 15m_{p}/16E^{*}r_{p}^{1/2}h_{1}^{5/2}.$$ where the final velocity of the particle rebound is *v*’*_y_*_1_ = (16*E^*^*·*r_p_*^1/2^·*h*_1_^5/2^/15*m_p_*)^1/2^. Therefore, the total contact time between a particle and the wall is available from solving the simultaneous Equations (17)–(19). During this contact process, the particle not only compress normally but also tangentially extrude the wall, which causes the material to protrude in one direction.

Through the overall analysis of the three processes, we get the expression of momentum change in the process of particle impact. Since the deformation can be recovered during the elastic indentation and recovery, the normal indentation depth *h* is approximately equal to the elastic-plastic indentation depth *h*_2_. Replace *h*_2_ with *h* in Equation (A8), the total indentation depth is,
(20)$$m_{p}v_{y0}^{2}-0.19\times \frac{R^{5/2}r_{p}^{1/2}\sigma_{y}^{5/2}}{{\left(E^{*}\right)}^{3/2}}=0.34\times \frac{\pi^{3}\sigma_{y}^{3}r_{p}^{2}h}{{\left(E^{*}\right)}^{2}}+\sigma_{y}\pi r_{p}^{}h_{}^{2}.$$

In this equation, the second term is the amount of particle velocity change in the elastic indentation process, which is approximately equal to 10^−6^~10^−8^ m·s^−1^. This velocity change is negligible compared to the initial particle impact velocity. Therefore, Equation (20) is recast and simplified as a relation between impact energy *E_p_* and indentation depth *h*, which is expressed by,
(21)$$E_{p}=Ah^{2}+Bh,$$ where $A=\frac{1}{2}\sigma_{y}\pi r_{p}^{}$, $B=0.17\times \frac{\pi^{3}\sigma_{y}^{3}r_{p}^{2}}{{\left(E^{*}\right)}^{2}}$.

Solve the monadic quadratic equation and take the positive solution to get the total indentation depth as shown below,
(22)$$h=\frac{-B+\sqrt{B^{2}-4AE_{p}}}{2A}.$$

### 2.2. Tangential Indentation under No Sliding Contact

Normally, a particle impacts the wall at an angle *α*, so depending on the angle of impact, two types of crater shapes under no sliding contact and sliding contact are shown in Figure 3. Figure 3a shows that a particle impacts the wall at a large angle *α*. The large normal pressure causes no slip contact between the particle and the wall. Due to continuous particle extrusion, the wall material is pushed in one direction to form a material lip. At this time, the shape of the crater is asymmetrical, and the tangential displacement of the particle contact point is *L*. Unlike the no sliding contact crater, the energy of the particles is dissipated in the sliding friction, resulting in a reduction in the indentation depth. As shown in Figure 3b, impact crater with sliding contact approximates symmetrical shape. Therefore, by comparing the maximum static friction force with the tangential contact force, it can be judged whether there is sliding or no sliding contact between the particle and the wall.

Based on Equation (A6), the maximum static friction of the wall can be expressed as,
(23)$$F_{x,\max}=\mu \cdot \left(0.17\times \frac{\pi^{3}\sigma_{y}^{3}r_{p}^{2}}{{\left(E^{*}\right)}^{2}}+\sigma_{y}\pi r_{p}^{}\delta_{y}\right).$$

By means of kinetic energy theorem, the relationship between the tangential velocity of a particle and the tangential force is,
(24)$$F_{x}t=m_{p}(v_{x0}-v_{x1}),$$ where *t* is the contact time of the indentation. When the final velocity in tangential direction is treated as 0 m·s^−1^ (i.e., *v_x_*_1_ = 0 m·s^−1^), Equation (25) is recast as,
(25)$$F_{x}=v_{x0}/\left[\sqrt{v_{y0}^{2}-\frac{16E^{*}r_{p}^{1/2}h_{1}^{5/2}}{15m_{p}}}\cdot \left(0.17\times \frac{\pi^{3}\sigma_{y}^{3}r_{p}^{2}h_{2}}{{\left(E^{*}\right)}^{2}}+\frac{1}{2}\sigma_{y}\pi r_{p}^{}h_{2}^{2}\right)\right].$$

Therefore, substituting the no sliding contact condition, i.e., *F_x_* ≦ *F_x,max_*, to Equations (24) and (25), one has,
(26)$$v_{x0}\leq \mu \sqrt{v_{y0}^{2}-\frac{16E^{*}r_{p}^{1/2}h_{1}^{5/2}}{15m_{p}}}\cdot {\left(0.17\times \frac{\pi^{3}\sigma_{y}^{3}r_{p}^{2}h_{2}}{{\left(E^{*}\right)}^{2}}+\frac{1}{2}\sigma_{y}\pi r_{p}^{}h_{2}^{2}\right)}^{2}.$$

Similar to the calculation of elastic-plastic contact time of normal indention (Equation (A8)), the relationship between the scratch length *L* and tangential particle velocity *v_x_* is expressed by,
(27)$$m_{p}v_{x0}^{2}-0.19\times \frac{R^{5/2}r_{p}^{1/2}\sigma_{x}^{5/2}}{{\left(E^{*}\right)}^{3/2}}=0.17\times \frac{\pi^{3}\sigma_{y}^{3}r_{p}^{2}L}{{\left(E^{*}\right)}^{2}}+\frac{1}{2}\sigma_{x}\pi r_{p}^{}L_{}^{2},$$
(28)$$AL^{2}+BL+C=0,$$ where $A=\frac{1}{2}\sigma_{y}\pi r_{p}^{}$, $B=0.17\times \frac{\pi^{3}\sigma_{y}^{3}r_{p}^{2}}{{\left(E^{*}\right)}^{2}}$, $C=-m_{p}v_{x0}^{2}$,
(29)$$L=\frac{-B+\sqrt{B^{2}-4AC}}{2A}.$$

According to geometry of an impact crater as shown in Figure 2, the volume of the eroded crater is expressed by *L* and *R*, and it is shown below,
(30)$$V_{s}=\frac{1}{3}(Lr_{p}^{1/2}h^{3/2}+r_{p}h^{2}).$$

### 2.3. Tangential Indentation under Sliding Contact

When the particles are in sliding contact with the wall, the contact areas can be divided into adhesive areas and sliding areas (Figure 4). The adhesive areas are mainly affected by normal extrusion force, and the range of sliding areas are controlled by tangential force transformed by particle kinetic energy. By means of Hertzian contact theory, the total tangential force of the adhesive and sliding areas is [21],
(31)$$\tau =\tau^{\left(1\right)}+\tau^{\left(2\right)},$$ where
$$\tau^{\left(1\right)}=\tau_{1}{\left(1-r_{p}^{2}/l^{2}\right)}^{1/2},$$
and
$$\tau^{\left(2\right)}=-\tau_{2}{\left(1-r_{p}^{2}/R^{2}\right)}^{1/2},$$ where *l* = *L*/2.

The relationship between stress and strain is shown below in the adhesive and sliding regimes,
(32)$$\delta_{x}=const\;r\;<\;R,$$
(33)$$\delta_{x}=\mu \cdot \sigma_{y}(r)\;R\;<\;r\;<\;l,$$

By means of Coulomb’s friction law, the shear stresses in the adhesive and sliding regimes can be treated as *τ*_1_ = *μσ*_0_ and *τ*_2_ = *μσ*_0_*R*/*l*, respectively. The total tangential force can be expressed by,
(34)$$F_{x}=\pi \left(\tau_{1}l^{2}-\tau_{2}R^{2}\right)=\frac{\pi \mu \sigma_{0}}{R}\left(l^{3}-R^{3}\right).$$

Simplifying Equation (A6) and ignoring the small amount in the first item, we obtained the relationship between normal and tangential contact forces in the adhesive regime according to *F_y_* = *σ_y_πR*^2^ and *σ*_0_ = *σ_y_*,
(35)$$\frac{R}{l}={\left(1-\frac{F_{x}}{\mu F_{y}}\right)}^{1/3}.$$

Therefore, the tangential maximum length of a crater is,
(36)$$L=2\sqrt{r_{p}h}/{\left(1-\frac{F_{x}}{\mu F_{y}}\right)}^{1/3},$$ where *F_x_* is obtained from Equation (26), and *F_y_* is obtained from Equation (A6). The depth of a cater caused by sliding contact is less than no sliding contact, but it is contrary for crater length. Therefore, the crater geometry under sliding contact is closer to the symmetrical vertebral body (Figure 3b). The volume of the eroded crater is expressed by,
(37)$$V_{s}=\frac{1}{3}Lr_{p}^{1/2}h^{3/2},$$

## 3. Results and Discussion

### 3.1. Particle–Wall Contact Time

Since particle contact time can influence the cutting length, the total contact times in two processes of particle pressing-in and rebound are first calculated. The contact time is calculated from Equations (17)–(19), and it is shown in Table 1.

According to calculated results, the particle–wall contact time is kept at the 10^−7^ s magnitude during the wall elastic indentation and rebound of the particle, but it is in the 10^−6^ s magnitude during the elastic-plastic indentation. This is because only limited energy of particle impact is absorbed in the complete elastic compression process. However, the vertical velocity of the particle only decreases by 10^−7^ m·s^−1^ and most impact energies are consumed during the elastic-plastic indentation. As a result, the elastic-plastic indentation process lasts the mostly when particle impact on the wall vertically, resulting in the approximately equivalence between elastic-plastic deformation and total deformation.

### 3.2. Deformation Volume of Metal Wall

(1) Normal indentation

The maximum indentation depths of target wall under different particle mass are calculated according to the Equation (22). Figure 5a shows an about 38.16 μm deep indentation is generated when a 10 g steel particle impacts on the Q235 steel wall at the velocity of 1 m·s^−1^. The indentation depth reaches 140.59 μm when the particle mass increases to 500 g. Therefore, the indentation depth is increased by 3.6 times when the particle mass is increased by 50 times given the constant impact velocity. When the particle velocity is increased 10 times (Figure 5b), the indentation depth is also increased by about 10 times. Therefore, the impact velocity of particles influences the indentation depth mostly for the same material.

Under the impact velocity of 10 m·s^−1^ (Figure 6), the indentation depth on the 35CrMo steel is 61.2% lower than that on the Q235 steel when the particle mass is 10 g. However, this value decreased to 60.3% when the particle mass reaches 500 g. It is generally maintained at about 60%, indicating the negative correlation between the yield strength (*σ_y_*) and indentation depth. Increasing material strength appropriately is conducive to resist deformation caused by particle impact. This is also the reason that harder wall surface (*HV* = *F*/*A_s_* = *F*/2*π*·*R*·*h*) has stronger resistance to erosion.

(2) Tangential deformation

Equation (26) shows the range of particle impact velocity for no sliding contact, where item 16*E^*^*·*r_p_*^1/2^·*h*_1_^5/2^/15*m_p_* is the velocity change in the elastic indentation process. According to the previous analysis results, this velocity change is much smaller than the initial particle impact velocity. Therefore, after ignoring a small amount, the following relationship is obtained.
(38)$$\cot\alpha =\frac{v_{x0}}{v_{y0}}\leq \mu \cdot {\left(0.17\times \frac{\pi^{3}\sigma_{y}^{3}r_{p}^{2}h_{2}}{{\left(E^{*}\right)}^{2}}+\frac{1}{2}\sigma_{y}\pi r_{p}^{}h_{2}^{2}\right)}^{2}.$$

The value of *π*^3^*σ_y_*^3^·*r_p_*^2^·*h*_2_/(*E^*^*)^2^ is only about 10^−7^ as well as the value of 0.5*σ_y_*^3^·*πr_p_*·*h*_2_^2^ is close to 10^−2^. Therefore, ignore the small amount to get the critical slip angle,
(39)$$\alpha_{1}=arc\cot\mu \cdot {\left(\frac{1}{2}\sigma_{y}\pi r_{p}^{}h_{}^{2}\right)}^{2}.$$

Table 2 shows the sliding angle for a 35CrMo wall under different particle impacts. The friction coefficient of metal surface is set to 0.22. According to calculation results, the critical impact angle for sliding (*α*_1_) decreases gradually with the increase of particle mass (or diameter). The critical angle is 86.8° when the particle mass is 10 g, indicating that no sliding contact only occurs when the impact angle ranges within 86.8°~90°. On the contrary, sliding contact occurs when the impact angle is smaller than 86.8°. Therefore, it is easier to cause sliding contact between the small-mass (small-diameter) particle and wall surface. The critical impact angle for sliding is smaller than 10.3° when the particle mass is higher than 100 g. Under this circumstance, no sliding contact occurs when the impact angle is between 10.3° and 90°, which indicates the particle with high mass mainly forms no sliding contact with the wall.

As for no sliding contact (Equation (25)), it is assumed that tangential momentum of the particle is completely dissipated in the process of wall deformation. But for sliding contact, particle still has certain momentum when it leaves the wall surface, indicating that part of the momentum is dissipated in the sliding contact. Grant [22] pointed out the relationship between normal or tangential velocity and the impact angle before and after the particle impacting on the wall surface, one has,
(40)$$\begin{array}{l} \lambda_{x}=v_{x1}^{}/v_{x0}^{}=0.993-1.76\alpha +1.56\alpha_{}^{2}-0.49\alpha_{}^{3}\\ \lambda_{y}=v_{y1}^{}/v_{y0}^{}=0.988-1.66\alpha_{}+2.11\alpha_{}^{2}-0.67\alpha_{}^{3} \end{array}$$

Changes of the tangential velocity have to be determined firstly to calculate the sliding contacting force based on the momentum theorem. Variations of normal and tangential velocity attenuation coefficients under different impact angles are shown in Figure 7. When the impact angle increases, the tangential attenuation coefficient decreases continuously. The maximum and minimum tangential attenuation coefficients are 0.96 (α = 1°) and 0.19 (α = 89°), indicating the negative correlation between tangential velocity and the impact angle. Additionally, the maximum and minimum normal attenuation coefficients are 0.98 (α = 89°) and 0.60 (α = 30°), which reflects that the maximum normal attenuation coefficient is achieved at about 30°. The normal attenuation coefficient is small in the range of α < 10° or α > 80°.

The normal and tangential attenuation coefficients are substituted into the Equation (26), thus getting the total tangential force on the wall,
(41)$$F_{x}=v_{x0}\left(1-\lambda_{x}\right)/\left[v_{y0}\left(1-\lambda_{y}\right)\cdot \left(0.17\times \frac{\pi^{3}\sigma_{y}^{3}r_{p}^{2}h_{}}{{\left(E^{*}\right)}^{2}}+\frac{1}{2}\sigma_{y}\pi r_{p}^{}h_{}^{2}\right)\right]$$

For example, a round steel ball (*m_p_* = 10 g, *d_p_* = 6.72 × 10^−3^ m) impacts on the 35CrMo steel surface at the velocity of 10 m·s^−1^, forming an indentation depth of 0.23 mm and a yield strength of 8.35 × 10^8^ Pa. Hence, it gets *F_y_* = *σ_y_πR*^2^ = *σ_y_πr_p_h* = 4052.41 N. All calculated tangential forces (*F_x_*) are shown in Figure 8. When the impact angle is between 1° and 10°, the tangential force and scratch length are decreased quickly. The tangential force decreases from about 600 N to lower than 100 N, while the scratch length decreases to lower than 3 mm. Due to the low occurrence frequency of small impact angle (<10°), the numerical values of tangential force and scratch length change slightly, which confirms the small influences of impact angle on scratch length.

After the scratch length is gained, according to the Equations (31) and (38), the erosion volumes of impact craters which are calculated by different masses of steel particle impacting on the 35CrMo wall at the velocity of 10 m·s^−1^ under no sliding and sliding contact. As shown in Table 3, the calculation results at the impact angle of 80° under no sliding contact (*v_x_* = 1.74 m·s^−1^, *v_y_* = 9.85 m·s^−1^) and impact angle of 10° under sliding contact (*v_x_* = 9.85 m·s^−1^, *v_y_* = 1.74 m·s^−1^) shows that the erosion volume under no sliding contact is significantly higher than that under sliding contact. Since the no sliding indentation depth is several times that of sliding indentation, the impact craters have greatly different areas. Although the tangential scratch length under sliding contact is longer than the no sliding length, the final volumes of impact craters under sliding contact is smaller than no sliding contact.

### 3.3. Gas-Solid Particle Impact Experiment

Applicability of the proposed model is verified by the gas-solid experimental system in Figure 9a. This system is composed of two air compressors, two flowmeters and filters, two particle tanks, and three cut-off valves. In the experiment, the air compressor 1 is open, the flow rate of the flowmeter 1 is adjusted, and the valve 1 is open for particle mixing erosion experiment. Particle backfilling is accomplished by supporting devices such as compressor 2. The erosion test section is shown in Figure 9b. In a closed space, sample clamp can adjust angle and height of the sample. A piece of 20 mm × 20 mm × 5 mm square P110 steel sample (*σ_y_* = 8.45 × 10^8^ Pa) is used in the experiment. The chemical composition and mechanical properties of P110 are shown in Table 4 and Table 5. The testing surface is grinded before the experiment by 300-mesh, 800-mesh, and 1200-mesh sand-paper to eliminate impurities and protect a smooth surface. Ceramic particles (*d_p_* = 0.6 mm), which are weighted of 100 g particles, are used in each experiment. The surfaces of samples were examined by scanning electron microscopy (SEM) (JSM-6390, JEOL. Co., Tokyo, Japan). In the independent impact region, sizes (radius *R*) of marginal impact craters on the P110 steel under different impact angles (30°, 45°, 60°, and 90°) and impact velocities (8 m·s^−1^, 12 m·s^−1^, 16 m·s^−1^, 20 m·s^−1^) are measured, and they were used to compared with theoretical calculation results. Since the calculated indentation depth (*h*) is in the micron size, the calculated critical impact angle for sliding is close to 90°. Hence, conditions for sliding particle contact are applied in the calculation.

In Figure 10, radius of the impact crater increases with the increase of impact velocity and angle. Specifically, the relationship between the radius of impact crater and the impact velocity conforms to the similar logarithmic function. The radius increment per unit change of impact speed is larger when the component impact velocity is higher. Influences of angle are more evident during changes of the high velocity flowing region. According to experimental results, relationships of radius of impact crater with flow velocity and impact angle conform to theoretical calculation results. However, the average relative errors under different impact angles are −15% (α = 90°), −12% (α = 60°), 8% (α = 45°), and 14% (α = 30°), which are attributed to the measurement and calculation errors. The relative error reaches the peak at 90°. Since sprayed particles from the nozzle change the moving direction by air flow on the sample surface, the actual impact angle is smaller than 90° although the included angle between the nozzle and sample is 90°. However, the impact angle in theoretical calculation is still determined as 90°, thus resulting in great error. In addition, particles are easy to slide under small impact angle, which determines the small value of experimental results.

## 4. Conclusions

Existing models only consider single factors of material deformation caused by single particle impact. To address this problem, a new elastic-plastic deformation prediction model of metal walls after single particle impact is constructed based on the Hertzian contact theory and conservation of momentum. Variations to the laws of the particle–wall contact force and contact time are discussed. Wall erosion volumes under indentation and cutting conditions of single particles are calculated. Additionally, the calculated results of the gas-solid jet experiments are compared.

Research results demonstrate that the elastic indentation time and elastic recovery time are similar in the three processes of contact, which are about 1/10 of the elastic-plastic contact time. Therefore, the elastic-plastic deformation volume is approximately equal to the total erosion volume in the impact process. The critical impact angle for sliding varies with particle diameters and masses. This critical angle determines the sliding condition between the particle and the wall. Larger particles are more difficult to slide during the contact and form a larger indentation depth and radius. Given the same impact conditions, the indentation length under sliding contact is several times that under no sliding contact, but the indentation depth is significantly smaller. According to particle impact experimental results, the relative error of the proposed model is smaller than 15% without involving too many empirical coefficients. The calculation error reaches the maximum at the angle of 90° and the minimum at 45°. Therefore, the proposed model performs better at 45° than at larger or smaller impact angles.

## Figures and Tables

**Figure 1 materials-12-00135-f001:**
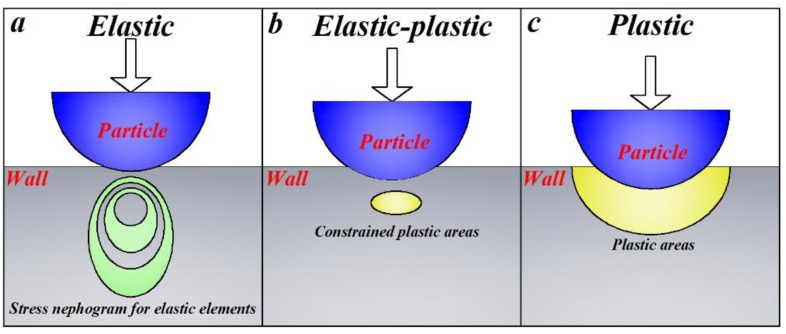
Schematic diagram of wall indentation caused by a rigid ball [19].

**Figure 2 materials-12-00135-f002:**
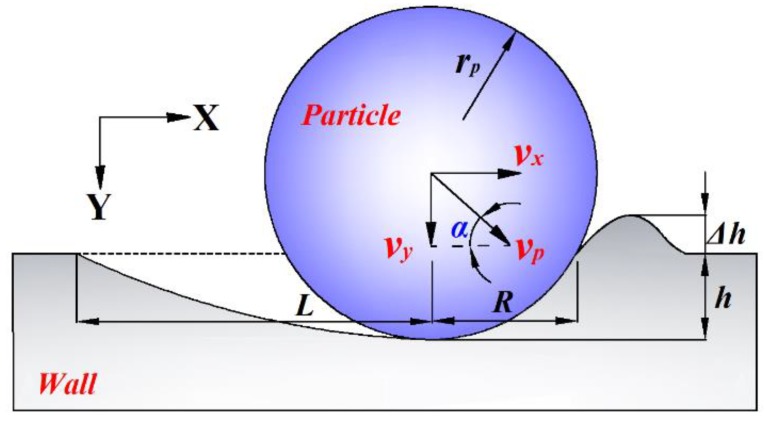
Schematic diagram of wall extrusion deformation caused by a particle impingement.

**Figure 3 materials-12-00135-f003:**
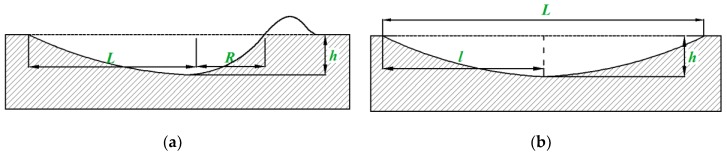
Schematic diagram of geometrical shape of particle impact crater: (**a**) no sliding contact; (**b**) sliding contact.

**Figure 4 materials-12-00135-f004:**
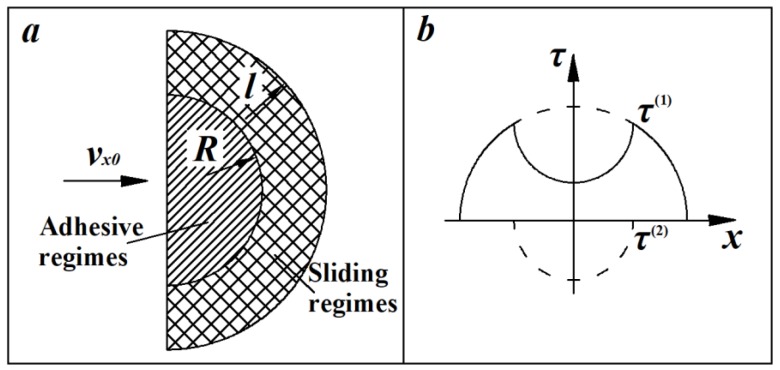
Sliding contact diagram: (**a**) adhesive areas and sliding areas; (**b**) the distribution for shear stress.

**Figure 5 materials-12-00135-f005:**
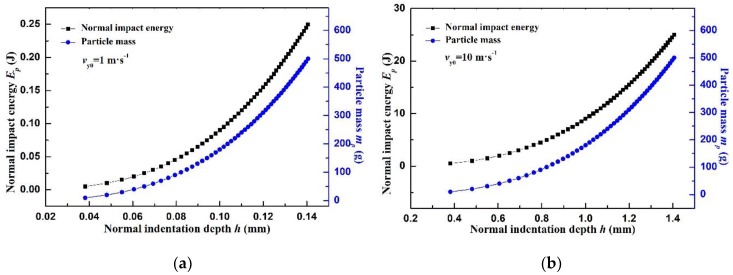
The impact energy and normal indentation depth versus different particle mass (Q235 for target wall): (**a**) normal impact velocity is 1 m/s; (**b**) normal impact velocity is 10 m/s.

**Figure 6 materials-12-00135-f006:**
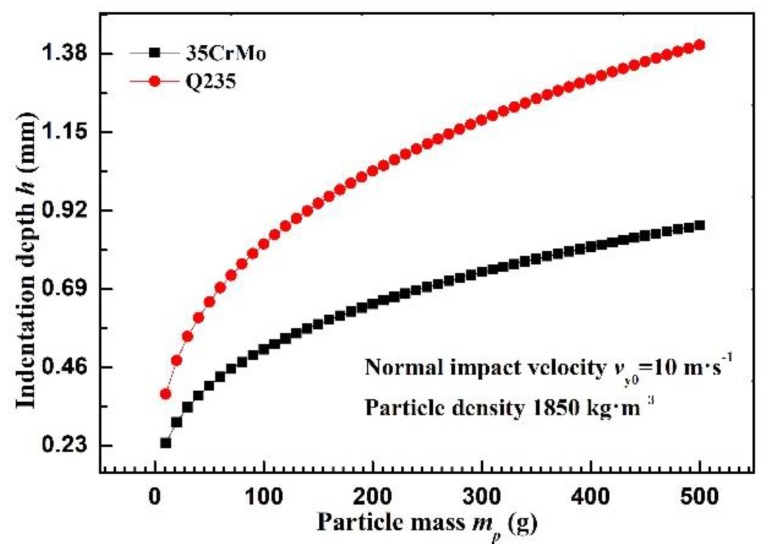
Indention depth versus particle mass for 35CrMo and Q235 wall.

**Figure 7 materials-12-00135-f007:**
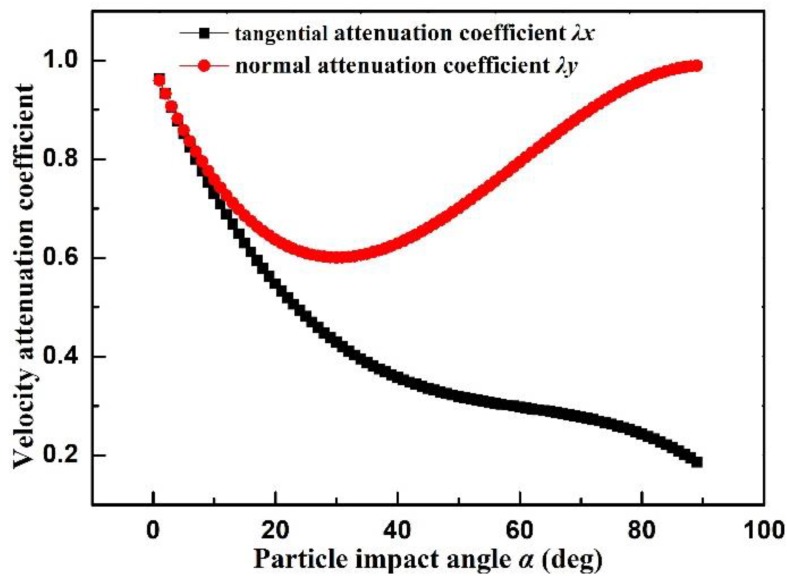
Velocity attenuation coefficient versus different impact angles.

**Figure 8 materials-12-00135-f008:**
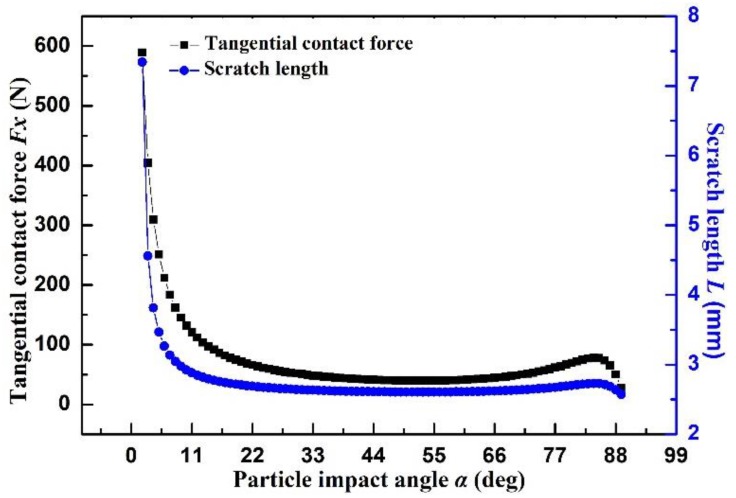
Tangential contact force and scratch length versus different impact angles.

**Figure 9 materials-12-00135-f009:**
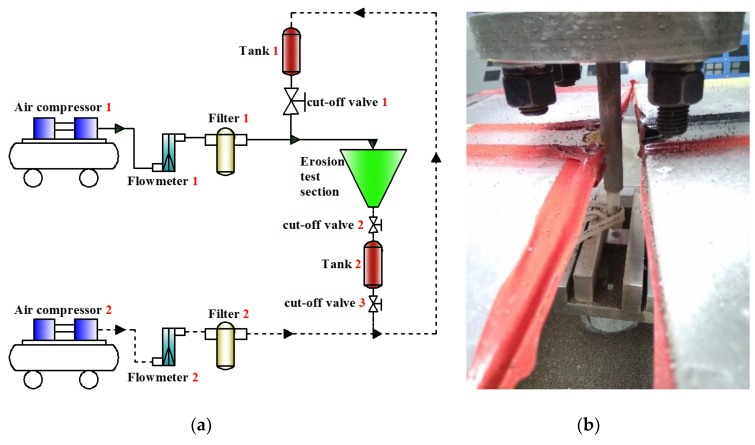
Gas-solid experimental system; (**a**) flow chart; (**b**) erosion test image.

**Figure 10 materials-12-00135-f010:**
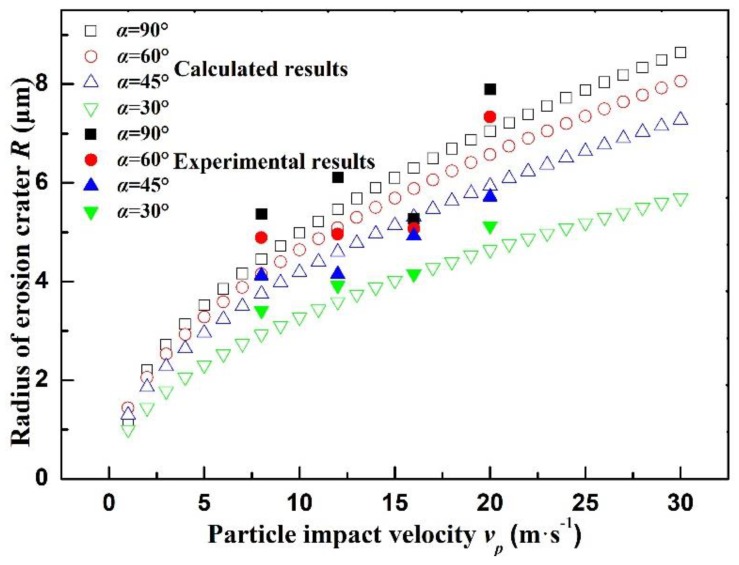
Comparison between calculated values and experimental values of crater radius.

**Table 1 materials-12-00135-t001:** The different contact times in three processes. (*v_y_*_0_ = 10 m·s^−1^, 35CrMo for target wall).

*m_p_*/g	*r_p_*/m	∆*v*/m·s^−1^	*h*_1_/mm	h_2_/mm	*t*_1_/s	*t*_2_/s	*t*_3_/s
10	6.73 × 10^−^^3^	1.033 × 10^−^^7^	2.111 × 10^−^^4^	0.238	4.928 × 10^−^^7^	2.002 × 10^−^^6^	4.829 × 10^−^^7^
100	1.45 × 10^−^^2^	1.033 × 10^−^^7^	4.548 × 10^−^^4^	0.513	4.928 × 10^−^^7^	2.002 × 10^−^^6^	4.829 × 10^−^^7^
200	1.83 × 10^−^^2^	1.033 × 10^−^^7^	5.730 × 10^−^^4^	0.646	4.928 × 10^−^^7^	2.002 × 10^−^^6^	4.829 × 10^−^^7^
300	2.09 × 10^−^^2^	1.033 × 10^−^^7^	6.559 × 10^−^^4^	0.739	4.928 × 10^−^^7^	2.002 × 10^−^^6^	4.829 × 10^−^^7^
400	2.30 × 10^−^^2^	1.033 × 10^−^^7^	7.220 × 10^−^^4^	0.814	4.928 × 10^−^^7^	2.002 × 10^−^^6^	4.829 × 10^−^^7^
500	2.48 × 10^−^^2^	1.033 × 10^−^^7^	7.777 × 10^−^^4^	0.877	4.928 × 10^−^^7^	2.002 × 10^−^^6^	4.829 × 10^−^^7^

**Table 2 materials-12-00135-t002:** The sliding angle for 35CrMo wall under different particle impacts. (*σ_y_* = 8.35 × 10^8^ Pa, *E^*^* = 2.34 × 10^11^ Pa).

*m_p_*/g	*r_p_*/m	*h*/m	*α*_1_/deg
10	6.726 × 10^−^^3^	2.380 × 10^−^^4^	86.8
20	8.474 × 10^−^^3^	2.998 × 10^−^^4^	77.6
30	9.701 × 10^−^^3^	3.432 × 10^−^^4^	63.7
40	1.068 × 10^−^^2^	3.778 × 10^−^^4^	48.7
50	1.150 × 10^−^^2^	4.070 × 10^−^^4^	36.0
60	1.222 × 10^−^^2^	4.325 × 10^−^^4^	26.8
70	1.287 × 10^−^^2^	4.553 × 10^−^^4^	20.4
80	1.345 × 10^−^^2^	4.760 × 10^−^^4^	15.8
90	1.399 × 10^−^^2^	4.950 × 10^−^^4^	12.6
100	1.449 × 10^−^^2^	5.127 × 10^−^^4^	10.3
110	1.496 × 10^−^^2^	5.293 × 10^−^^4^	8.5
120	1.540 × 10^−^^2^	5.449 × 10^−^^4^	7.2
130	1.581 × 10^−^^2^	5.596 × 10^−^^4^	6.1
140	1.621 × 10^−^^2^	5.736 × 10^−^^4^	5.3
150	1.659 × 10^−^^2^	5.869 × 10^−^^4^	4.6
160	1.695 × 10^−^^2^	5.997 × 10^−^^4^	4.0
170	1.729 × 10^−^^2^	6.119 × 10^−^^4^	3.6
180	1.763 × 10^−^^2^	6.237 × 10^−^^4^	3.2
190	1.795 × 10^−^^2^	6.350 × 10^−^^4^	2.8
200	1.826 × 10^−^^2^	6.460 × 10^−^^4^	2.6

**Table 3 materials-12-00135-t003:** Geometric size of impact craters with sliding contact or no sliding contact.

Particle Parameters		No Sliding Contact			Sliding Contact		
*m_p_*/g	*r_p_*/m	*h*/mm	*L*/mm	*V_s_*/mm^3^	*h*/mm	*L*/mm	*V_s_*/mm^3^
10	6.726 × 10^−^^3^	0.234	0.058	3.868	0.041	2.651	1.905
20	8.474 × 10^−^^3^	0.295	0.074	11.717	0.052	2.152	2.455
30	9.701 × 10^−^^3^	0.338	0.084	17.575	0.060	1.701	2.543
40	1.068 × 10^−^^2^	0.372	0.093	23.433	0.066	1.739	3.149
50	1.150 × 10^−^^2^	0.401	0.100	29.291	0.071	1.816	3.814
60	1.222 × 10^−^^2^	0.426	0.106	35.150	0.075	1.497	4.501
70	1.287 × 10^−^^2^	0.448	0.112	41.008	0.079	1.277	5.197
80	1.345 × 10^−^^2^	0.469	0.117	46.866	0.083	0.753	5.899
90	1.399 × 10^−^^2^	0.488	0.122	52.725	0.086	0.425	6.605
100	1.449 × 10^−^^2^	0.505	0.126	58.583	0.089	0.193	7.313

**Table 4 materials-12-00135-t004:** Chemical composition of P110.

Material	C	Si	Mn	Cr	P	S	Ni	Mo	Cu
P110	0.28	0.26	1.68	0.03	0.013	0.0013	0.04	0.03	0.044

**Table 5 materials-12-00135-t005:** Mechanical properties of P110.

Material	Tensile Strength/Mpa	Yield Strength/Mpa	Elongation/%	Hardness/HV
P110	845	940	25	299

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
