# Peer review of "A New Phenomenological Model for Single Particle Erosion of Plastic Materials"

_materials, 2019, doi:10.3390/ma12010135_

Reviewer 1 Report

The study focuses on a relevant topic, and should be considered for publication.

However, the weakness mostly relates to the structure of the document, and requires revision.

In particular, the document is in general very long and has the structure/content of a research report, rather than a scientific journal paper.

The long list of equations (54 in total!) also reflects this statement.

The authors are hence warmly encouraged to consistently reduce the size of the paper, moving towards a concise document.

The full paper will take benefit from this corrections, helping the authors to better emphasise their research studies and the novel contributions, compared to existing literature investigations.

In doin so, please consider also to reduce the number of equations, removing part of them or at least post-poning some of them in a final annex. This will facilitate the readability of the document.

Finally, re-check also the numbering of equations (what happens after equation (49)?)

And extend the list of references, so as to achieve an appropriate number of literature documents (>20) in support of this scientific contribution.

Author Response

Comments and Suggestions for Authors

The study focuses on a relevant topic, and should be considered for publication. However, the weakness mostly relates to the structure of the document, and requires revision.

Reply: Thank you very much for your affirmation of our article. This article has more descriptions of model establishment because it is part of the erosion-corrosion prediction system, including erosion calculation, corrosion measurement and erosion-corrosion analysis in the future. I am sorry for the formatting error that appeared in the article and have corrected it. About the seven questions you raised, the first five questions I answered in the form of a list.

1.In particular, the document is in general very long and has the structure/content of a research report, rather than a scientific journal paper.

2.The long list of equations (54 in total!) also reflects this statement.

3.The authors are hence warmly encouraged to consistently reduce the size of the paper, moving towards a concise document.

4.The full paper will take benefit from this corrections, helping the authors to better emphasise their research studies and the novel contributions, compared to existing literature investigations.

5.In doin so, please consider also to reduce the number of equations, removing part of them or at least post-poning some of them in a final annex. This will facilitate the readability of the document.

Reply: Thank you for your advice. We list the specific deletions as shown in the table below.

Content

Before deletion

After deletion

Remarks

page

21

18

Remove part of the duplicate narrative.

equation

54

42

Some of the formulas are listed in Appendix B.

figure

14

10

Experiments are briefly described.

Since the main research in this paper involves a detailed model building process, more equations and tables will appear. In the case of ensuring complete content, we try our best to delete unimportant text.

6.Finally, re-check also the numbering of equations (what happens after equation (49)?)

Reply: I'm sorry for this format error. We have corrected the numbering of equations and checked other parts. Thank you very much.

7.And extend the list of references, so as to achieve an appropriate number of literature documents (>20) in support of this scientific contribution.

Reply: According to your advice, we have added three references (Refs.16,17,18) in the introduction section. The establishment of semi-empirical models, which are used to compared with the new model, is detailed in these references. Thank you very much.

Reviewer 2 Report

This paper presents an interesting mechanistic model to predict the contact time and volume loss due to impact of a single particle based on Hertzian theory. The model also takes into account the sliding and no sliding contacts. The model derivation and results are very interesting and the paper worthy of publications. There are a few comments which can improve the quality of the paper.

1. In section 2 (model establishment) – paragraph 2 – Fig 2 (a) should be Fig 1 (a)

2. In the abstract authors claimed that no empirical coefficients were used in the model. However, the Grant and Tabakof coefficient restitution correlation has been used in the model which is based on experimental data. The last sentence of the abstract should be modified.

3. There is a slip velocity between gas and particles, therefore particle speeds should be measured. How the particle impact speeds were measured in the tests? Did you use Particle Image Velocimetry (PIV)? What technique did you use?

4. What material was used for the erosion test as samples? And what material properties were used in the calculations to compare with the experimental data?

5. Due to difficulties in the measurements of the single particle crater, if authors find any numerical results (Finite Element) and compare with their model, it can validate the model better.

Author Response

Comments and Suggestions for Authors

This paper presents an interesting mechanistic model to predict the contact time and volume loss due to impact of a single particle based on Hertzian theory. The model also takes into account the sliding and no sliding contacts. The model derivation and results are very interesting and the paper worthy of publications. There are a few comments which can improve the quality of the paper.

Reply: Thank you for your praise of our work. Thanks for your suggestion and advice on our paper. We have revised the manuscript according to your detailed suggestions.

1. In section 2 (model establishment) – paragraph 2 – Fig 2 (a) should be Fig 1 (a)

Reply: I'm sorry for this format error. I have corrected this error and checked other similar points. Thank you very much.

2. In the abstract authors claimed that no empirical coefficients were used in the model. However, the Grant and Tabakof coefficient restitution correlation has been used in the model which is based on experimental data. The last sentence of the abstract should be modified.

Reply: Thank you very much. As you said, the Grant and Tabakof coefficient does appear in the article. Therefore, the no empirical coefficient has been changed to too many empirical coefficients in the abstract and introduction.

3. There is a slip velocity between gas and particles, therefore particle speeds should be measured. How the particle impact speeds were measured in the tests? Did you use Particle Image Velocimetry (PIV)? What technique did you use?

Reply: Yes, indeed, there is a slip velocity between gas and particles. The Stokes number is a dimensionless number characterizing the behavior of particles suspended in a fluid flow, and it can be expressed by

 The calculated results show that the Stokes numbers are greater than 2000. Therefore, there is a significant speed difference between particles and gas in this experiment. Instead of using PIV equipment to measure particle velocity, we use a high-speed camera to capture the particle position at different times, and then calculate the particle velocity. However, some problems were encountered during the test. For example, when testing the particle nozzle outlet particles, the instrument will be collided by high-speed particles. So first, the particle velocity near the nozzle section is measured, and then the gas valve is adjusted to meet the required particle velocity.

4. What material was used for the erosion test as samples? And what material properties were used in the calculations to compare with the experimental data?

Reply: The metal used in this experiment is P110, which is usually used as the tubing or casing materials in oil field. The chemical composition and mechanical properties of P110 are shown in Tables 4 and 5.

5. Due to difficulties in the measurements of the single particle crater, if authors find any numerical results (Finite Element) and compare with their model, it can validate the model better.

Reply: Thank you for your advice. There are many studies focus on the calculation of erosion crater using finite element method (FEM) such as the following figure. In this paper, we attempt to establish a simple and convenient algorithm for predicting industrial erosion. Therefore, most of the content is focused on model building and results discussion. As another reviewer said, the document is in general very long and has the structure/content of a research report. Therefore, an experiment was carried out to compare the results without detailed comparison of the finite element results. In future studies, we may consider comparing the calculated results with the finite element results. Thank you very much.

 Wang Y F , Yang Z G . Finite element model of erosive wear on ductile and brittle materials. Wear, 2008, 265(5):871-878.

Round  2

Reviewer 1 Report

Accepted